# A Randomized Controlled Trial Evaluating the Effects of a Probiotic Containing *Lactobacillus helveticus* R0052 and *Bifidobacterium longum* R0175 on Gastrointestinal Symptoms and Metabolomic Profiles in Female Dancers

**DOI:** 10.3390/ijms26125823

**Published:** 2025-06-18

**Authors:** Jakub Wiącek, Karolina Skonieczna-Żydecka, Igor Łoniewski, Daniel Styburski, Mariusz Kaczmarczyk, Joanna Karolkiewicz

**Affiliations:** 1Department of Food and Nutrition, Poznan University of Physical Education, 61-871 Poznan, Poland; wiacek@awf.poznan.pl (J.W.); karolkiewicz@awf.poznan.pl (J.K.); 2Department of Biochemical Research, Pomeranian Medical University, 70-204 Szczecin, Poland; daniel.styburski@sanprobi.pl (D.S.); mariusz.kaczmarczyk@pum.edu.pl (M.K.); 3Sanprobi Sp. z o.o. Sp.K., 70-535 Szczecin, Poland; sanprobi@sanprobi.pl

**Keywords:** gut metabolome, probiotics, sports nutrition, *Lactobacillus helveticus* R0052, *Bifidobacterium longum* R0175

## Abstract

Dancers experience physical and psychological stressors that can impact gut health. We hypothesized that a three-month supplementation with *Lactobacillus helveticus* R0052 and *Bifidobacterium longum* R0175 would result in measurable alterations in the fecal metabolomic profile and improve gastrointestinal symptomatology in dancers. Of the 51 volunteers, 26 female dancers were randomized to a 12-week trial (NCT05567653). A homogenous group of 16 (probiotic: *n* = 5; placebo: *n* = 11) was analyzed. The participants received *L. helveticus* R0052 and *B. longum* R0175 (3 × 10^9^ colony-forming units/day) or a placebo. Baseline dietary intake and body composition were recorded. Fecal samples were analyzed using liquid chromatography–mass spectrometry, and gastrointestinal symptoms were assessed with the Rome IV questionnaire. Statistical methods included principal component analysis, mixed-effects models, and analysis of variance–simultaneous component analysis (ASCA). The study revealed shifts in the probiotic group’s fecal metabolome (permutation test *p* = 0.026), including a reduction in (2RS)-2-(4-hydroxyphenyl)propionic acid (*p* = 0.023). No improvement in gastrointestinal symptoms was observed. No adverse events occurred. *L. helveticus* R0052 and *B. longum* R0175 may alter the gut metabolome, notably (2RS)-2-(4-hydroxyphenyl)propionic acid, but small sample size and absent symptom improvement limit the conclusions. Larger studies with varied doses and blood metabolite analysis are needed to confirm relevance.

## 1. Introduction

Dancers experience intense pressure to maintain a specific physique, often adopting restrictive eating habits that increase the risk of undernutrition and gastrointestinal disorders, while late-night rehearsals disrupt circadian rhythms, impairing sleep and recovery, and frequent travel exposes them to diverse environments and pathogens, heightening infection risk [1,2,3,4,5]. Comprehensive nutritional support is essential to help dancers manage these challenges effectively. Similarly to athletes, dancers inherently experience conditions linked to significant gut microbiota and metabolomic changes, as their high-intensity, prolonged training—often exceeding 30 min multiple times per week—falls within the exercise parameters known to alter microbial composition and metabolism [6,7]. The physical demands of dance, combining aerobic and anaerobic elements, drive metabolic shifts in energy metabolism, oxidative stress, and recovery, while long-term training fosters sustained metabolomic changes, promoting such metabolites as short-chain fatty acids (SCFAs), amino acid derivatives, and lipid mediators linked to energy efficiency, anti-inflammatory responses, and neuromuscular adaptation [8,9]. The ability of dancers to maintain these rigorous physical demands positions them as an ideal population for exploring the interplay between exercise, the gut microbiota, and the metabolome.

Probiotic supplementation has been shown to enhance athletic performance by modulating the gut microbiota and the metabolome, increasing beneficial metabolites such as SCFAs and reducing inflammation and oxidative stress [10]. This modulation may lead to increased production of short-chain fatty acids (SCFAs) such as acetate, propionate, and butyrate. SCFAs play a crucial role in maintaining the intestinal barrier integrity by serving as energy sources for colonocytes and promoting the expression of tight junction proteins. Additionally, SCFAs exhibit immunomodulatory effects; for instance, butyrate has been shown to promote the differentiation of regulatory T cells (Tregs) and suppress pro-inflammatory cytokine production, contributing to immune homeostasis [11]. In the context of physical performance, these mechanisms are particularly relevant. Enhanced intestinal barrier function can reduce the translocation of endotoxins, thereby decreasing systemic inflammation—a common issue in athletes undergoing intense training. Moreover, the immunomodulatory effects of SCFAs can support recovery and reduce the incidence of infections, which are prevalent among athletes due to physical stress [12,13]. Specific strains, such as those from the *Lactobacillus* and *Bifidobacterium* genera, have been shown to support endurance, reduce fatigue, and enhance nutrient absorption, including branched-chain amino acids and iron, thereby aiding muscle repair and aerobic capacity [14,15,16]. *Lactobacillus helveticus* R0052 and *Bifidobacterium longum* R0175 are among the most frequently studied probiotics affecting mental health and sleep quality [17,18,19]. Moreover, they were shown to produce a range of metabolites, such as SCFAs, gamma-aminobutyric acid (GABA), and cytokines, which support gut microbiota stability, enhance the intestinal barrier function, and modulate immune responses, all of which are critical for athletes exposed to high physical and metabolic demands [20,21]. *L. helveticus* R0052 and *B. longum* R0175 have been shown to restore the gut barrier integrity and reduce systemic inflammation in stress-induced models. For example, this probiotic combination significantly reversed intestinal hyperpermeability and lowered inflammatory markers in a rat model of myocardial infarction-induced depression [22]. A study involving Syrian hamsters subjected to social defeat stress found that administration of *L. helveticus* R0052 and *B. longum* R0175 altered the gut microbial composition and increased levels of anti-inflammatory cytokines such as interleukin-4 (IL-4), interleukin-5 (IL-5), and interleukin-10 (IL-10) [23]. Various subspecies of *Lactobacillus helveticus* and *Bifidobacterium longum* have been studied for their potential roles in maintaining the gut barrier integrity and modulating inflammation. For example, analysis of the cell wall structure of *Lactobacillus helveticus* using atomic force microscopy revealed the presence of twisted peptidoglycan fibers approximately 26 nm in thickness, which may contribute to maintaining the bacterium’s structural integrity and potentially influence its interactions with the intestinal epithelium [24]. Additionally, *Bifidobacterium longum* subsp. *infantis* FJSYZ1M3 has been shown to ameliorate dextran sulfate sodium (DSS)-induced colitis in mice by maintaining the intestinal barrier, regulating inflammatory cytokines, and modifying the gut microbiota. Specifically, this strain improved the integrity of intestinal tight junctions, relieved mucus layer damage, and inhibited epithelial cell apoptosis, thereby maintaining the intestinal barrier. Additionally, it significantly affected the levels of inflammatory cytokines interleukin-6 (IL-6), interleukin-1β (IL-1β), and IL-10 in the colon, thus relieving inflammation in colitis mice [25].

This study hypothesized that a three-month-long supplementation with a probiotic containing *Lactobacillus helveticus* (Rosell-0052) and *Bifidobacterium longum* (Rosell-0175) would lead to significant changes in the fecal metabolomic profile of dancers compared to those receiving a placebo. While dancers do not undergo training as rigorously as elite athletes such as marathon runners or rugby players, the demands of their physical activity align with the study’s goals. The research focused exclusively on female participants, as their enrollment rate was 20 times higher than that of males. Moreover, probiotic effects might differ based on sex, and including both genders could have introduced variability, making data interpretation more complex [26]. The primary outcomes included the assessment of untargeted metabolomic shifts and the levels of specific metabolites potentially influenced by the intervention. Untargeted metabolomics enabled broad comparisons between groups, identifying metabolic shifts and potential early perturbations without prior assumptions. As a secondary outcome, we evaluated the impact of probiotic supplementation on gastrointestinal symptoms, including constipation, diarrhea, and sensations of fullness, measured using the Rome IV questionnaire. We hypothesized that probiotic supplementation would reduce Irritable Bowel Syndrome (IBS)-like symptoms, such as abdominal pain and constipation.

## 2. Results

### 2.1. Baseline Characteristics

Based on the baseline data, no significant differences were observed between the placebo and probiotic groups in terms of anthropometric variables, including age, body mass, BMI (body mass index), body fat percentage, and physical activity levels. Similarly, the dietary characteristics of the participants were comparable between the two groups at the start of the study. Parameters such as total energy intake, proteins, fats, total cholesterol, carbohydrates, sucrose, fiber, and the 14-Item Mediterranean Diet Assessment Tool showed no statistically significant variations. The characteristics of the groups and the diet are presented in Table 1. The participant data were previously used in a separate publication focusing on probiotic effects unrelated to the outcomes reported here.

These findings indicate that the groups were homogenous at baseline, providing a solid foundation for assessing the effects of the probiotic intervention without confounding factors related to the initial differences in body composition, physical activity, or dietary habits. Although overall adherence to the Mediterranean diet did not differ substantially between the groups, the estimated intake levels of specific food groups within this pattern showed greater variability. Despite the lack of statistically significant differences based on the Mann–Whitney U test, certain trends, particularly in the consumption of legumes and vegetables, which are known to significantly influence the gut microbiota composition, suggest that habitual intake may not have been entirely uniform across the participants.

### 2.2. Untargeted Metabolomics (Liquid Chromatography–Mass Spectrometry)

Principal component analysis (PCA) did not reveal a distinct clustering of the samples by intervention group or time point (Figure 1), indicating broadly overlapping global metabolomic profiles at baseline and post-intervention. Nonetheless, minor differences, potentially attributable to unreported or low-resolution dietary variation, cannot be excluded as contributors to specific metabolite-level changes, despite efforts to evaluate dietary adherence and comparability.

A heatmap with log-transformed values visualized metabolic differences, with metabolites as rows and individuals as columns. The difference in concentrations between the probiotic group (red) and the placebo group (blue) may indicate an intervention-related shift; however, this does not imply a unidirectional effect (Figure 2). Metabolites were clustered by similarity, with annotations for group (A—placebo, B—probiotic) and time (0—pre, 1—post). The metabolite list is in the Appendix A. A permutation test confirmed a significant intervention effect (*p* = 0.026), while time and interaction effects were non-significant, indicating the intervention as the primary driver of change.

A mixed-effects model was applied to evaluate whether the trajectories of metabolite changes over time varied between the two intervention groups. The analysis revealed that one metabolite, (2RS)-2-(4-hydroxyphenyl)propionic acid, demonstrated the strongest statistical association, with an uncorrected *p*-value of 0.023, based on the Kenward–Roger approximation. These findings are visualized in Figure 3A. The predictor effect plot illustrates the coefficients for the fixed effects, including intervention, time, and their interaction. The significant interaction observed appears to be driven predominantly by baseline differences between the groups. In addition to the mixed-effects model, a multivariate empirical Bayes analysis of variance (MEBA) was performed (Figure 3B) to assess temporal profiles across biological conditions. MEBA identified (2RS)-2-(4-hydroxyphenyl)propionic acid as one of the metabolites with the highest differential time-dependent response between the intervention groups.

The ASCA (ANOVA–simultaneous component analysis) identified four metabolites as “well-modeled,” indicating their strong contribution to defining the metabolic differences between the groups. Among these, (2RS)-2-(4-hydroxyphenyl)propionic acid stood out as a key metabolite linked to the intervention, as highlighted by their significance in the mixed-effects model. These findings, together with observations from the heatmap and fold change analyses (Table 2), suggest that the intervention induces distinct metabolic shifts. Only metabolites with a log_2_ fold change greater than 2 were selected from the cross-sectional analysis, corresponding to at least a fourfold increase in the probiotic group.

Cross-sectional analysis at the endpoint revealed substantial changes in metabolite levels following probiotic supplementation, with several compounds, including hippuric acid and curcumin, showing marked increases (log_2_FC > 2). However, not all fold changes reached statistical significance due to interindividual variability. Some of the identified metabolites, such as curcumin and naringenin, may reflect the intake of phytochemical-rich supplements; however, none of the participants reported the use of such supplements in the relevant section of the dietary records, and adherence to the Mediterranean diet pattern, used to assess dietary similarity between the groups, did not reveal marked differences. To further assess the robustness of these findings, a *t*-test (Table 3) was performed, identifying a subset of metabolites that exhibited statistically significant differences between the probiotic and placebo groups.

An FDR of ~0.46 indicates that after correction, none of the metabolites reached statistical significance. However, this does not exclude biologically relevant effects, as FDR correction is highly conservative in small sample sizes and can mask meaningful trends, especially in exploratory studies.

### 2.3. Gastrointestinal Complaints (Rome-IV Questionnaire)

Gastrointestinal symptom assessment showed a general reduction in both groups, though most changes were not statistically significant. As shown in Table 4, only constipation showed a significant time effect (*p* = 0.0298). No significant effects were found for postprandial fullness, abdominal pain, menstrual pain, or diarrhea.

## 3. Discussion

This is the first study to evaluate the untargeted gut metabolome in artistic athletes following a 3-month probiotic supplementation period. The present study shows that participation in a 3-month study involving a probiotic formula consisting of *Lactobacillus helveticus* R0052 and *Bifidobacterium longum* R0175 may affect the gut metabolome of professional dancers. Research on the metabolomic effects of *L. helveticus* R0052 and *B. longum* R0175 is limited, but some evidence supports their influence. A study using the SHIME^®^ (Simulator of the Human Intestinal Microbial Ecosystem, ProDigest BV, a spin-off from Ghent University, Ghent, Belgium) model in adults with mild anxiety found that a two-week supplementation increased acetic acid and total SCFAs while modulating the gut microbiota, notably boosting *Lactobacillus* and *Olsenella* and reducing *Lachnospira* and *Escherichia*/*Shigella* [27].

The key metabolite 2-(4-hydroxyphenyl)propionic acid (2-HPP), identified in our analysis, is involved in the gut microbial metabolism of isoflavonoids such as genistein and daidzein from soy. According to the Human Metabolome Database, 2-HPP is present in several biofluids, including blood and urine [28]. Probiotics can modulate the gut microbiota and influence the metabolism of plant-derived antioxidants by affecting microbial enzyme activity, such as the β-glucuronidases involved in isoflavone deconjugation [29]. This may enhance alternative pathways and increase the production of 2-(4-hydroxyphenyl)propionic acid (2-HPP), a gut-derived metabolite of genistein from soy [30]. Along with dihydrogenistein, 2-HPP is produced via microbial cleavage of genistein’s isoflavonoid structure, indicating active microbial metabolism and subsequent absorption [31]. For example, the bacterial strain SY8519 can convert daidzein into O-desmethylangolensin and produce optically active 2-HPP from genistein, with chiral HPLC confirming the R-enantiomer. In adipocyte studies, both genistein and 2-HPP inhibited leptin secretion, though 2-HPP was less potent [32]. 2-HPP has been associated with effects on lipid metabolism, including the inhibition of lipid accumulation and improvement of lipid processing [33]. The anaerobic gut bacterium *Eubacterium ramulus*, commonly present in the human gut, metabolizes flavonoids such as genistein and daidzein into 2-HPP and O-desmethylangolensin by breaking down flavonoid rings into hydroxyphenylpropionic and hydroxyphenylacetic acids, along with acetate and butyrate [34]. Its flavonoid-degrading activity has been confirmed both in vitro and in vivo, including in rats fed quercetin, where 3,4-dihydroxyphenylacetic acid was detected in feces. Probiotic-driven microbiome enrichment may support the colonization of *E. ramulus*, as flavonoid-rich diets are known to increase its abundance [35]. Studies on isoflavone and lignan metabolism show that gut microbiota-mediated deglycosylation increases flavonoid aglycones (e.g., quercetin, kaempferol), enhancing their bioavailability and activity. A decline in antioxidant activity and phenolic content, alongside elevated levels of 3,4-dihydroxyphenylacetic acid, 2-HPP, protocatechuic acid, and catechol, indicates partial or full flavonoid catabolism by the microbiota [36]. Moreover, 2-HPP has been linked to increased *Bifidobacterium* abundance in rats, and long-term intake of phenolic compounds suggests 2-HPP may contribute to gut microbiota modulation and adaptation [37]. The reduced 2-HPP levels in the probiotic group may reflect enhanced intestinal absorption or increased microbial conversion into secondary metabolites, both influenced by probiotic-driven changes in the gut microbiota. These results support the hypothesis that probiotics significantly alter the gut metabolome, particularly in pathways linked to plant antioxidant metabolism. To date, no studies have directly examined 2-HPP in the context of athletic performance or health. While specific studies on 2-HPP in athletic populations are lacking, the general benefits of polyphenol metabolites on exercise performance and recovery have been documented. For instance, polyphenol supplementation has been associated with improved aerobic endurance and fat oxidation in athletes [38].

The magnitude of fold change (FC)—in some cases exceeding several orders—indicates a robust biological response to probiotic supplementation. Notably, hippuric acid levels increased markedly in the probiotic group post-intervention. Hippuric acid is a well-established host microbiota co-metabolite, produced through the microbial metabolism of dietary polyphenols and aromatic amino acids [39]. Elevated hippuric acid levels are commonly linked to enhanced microbial activity and polyphenol-rich diets, correlating with greater microbiota diversity [40]. Our results may indicate that the intervention modulated microbial pathways associated with hippurate production, possibly by enhancing the biotransformation of dietary polyphenols or aromatic amino acids such as phenylalanine. Changes also affected polyphenol-related and diet-derived metabolites closely linked to gut microbiota activity [41]. Naringenin and genistein levels increased markedly in the probiotic group after intervention. These findings may suggest enhanced microbial release and bioavailability potentially influenced by probiotic activity, an effect that was not evident in the placebo group. However, this observation may also reflect uncontrolled intake of specific dietary components, as the participants were instructed—but not strictly monitored—to maintain identical dietary habits prior to the first and second sampling time points.

The probiotic intervention may have modulated the bile acid profile, as indicated by lower endpoint levels of certain bile acid derivatives—such as hyocholic acid—in the probiotic group. This aligns with known probiotic activity, particularly bile salt hydrolase (BSH)-mediated deconjugation, which reduces conjugated bile acids while releasing free bile acids and amino acids such as taurine or glycine [42,43]. Furthermore, bile acids, including HCA, interact with host receptors such as the farnesoid X receptor (FXR) and G protein-coupled bile acid receptor 1 (TGR5), which play roles in regulating inflammation and maintaining the intestinal barrier integrity [44]. The observed reductions in tryptophan and phenylalanine levels following probiotic supplementation may result from enhanced microbial metabolism of these aromatic amino acids. Probiotics can modulate the gut microbiota composition, leading to increased catabolism of tryptophan via the kynurenine pathway and phenylalanine through various microbial enzymatic activities. This enhanced microbial utilization could decrease the systemic availability of these amino acids, as supported by studies indicating that probiotic intervention influences aromatic amino acid metabolism [45,46] and our observation regarding HCA levels.

In our study, we did not observe significant improvements in gastrointestinal symptoms such as abdominal pain or bloating following probiotic supplementation, as assessed by the Rome IV criteria. The absence of symptom improvement may reflect low baseline GI symptom severity (e.g., compared to marathon runners or IBD patients) or inaccuracies in dietary reporting during the intervention. However, a reduction in constipation frequency was noted across both groups, which may reflect not only a placebo effect but also the impact of increased self-monitoring and adherence to study-related behavioral recommendations, such as maintaining hydration, regular meal timing, or reduced training load prior to sample collection [47]. Although probiotics are considered a promising intervention for functional gastrointestinal disorders such as IBS—with some clinical trials and meta-analyses reporting symptom relief, particularly in Rome IV-diagnosed patients—these findings remain inconsistent due to considerable heterogeneity and strain-specific effects [48,49]. In light of previous preclinical findings, it is noteworthy that the combination of *L. helveticus* R0052 and *B. longum* R0175 has been shown to attenuate visceral hypersensitivity and normalize the hypothalamic–pituitary–adrenal (HPA) axis activity more effectively than either strain alone, suggesting a synergistic effect in regulating stress-induced alterations in gut–brain signaling [50].

### Limitations

This study has limitations that should be considered. The small sample size, largely due to difficulties in assembling homogeneous groups, may reduce generalizability. Although we attempted to recruit participants from various dance academies, performance ensembles, and theaters, practical constraints and a low number of male volunteers ultimately hindered us from achieving the intended sample size. However, the strict exclusion criteria improved group uniformity and minimized confounding. As the study sample consisted exclusively of female participants, the results cannot be extrapolated to male populations. Considering the potential sex-based differences in both probiotic responsiveness and endocannabinoid system function, future research should aim to include male participants to examine possible divergences in outcomes. Although recruitment efforts extended across several dance academies, performance ensembles, and theater groups, practical limitations and a lack of male volunteers made it impossible to achieve a sex-balanced cohort. The lack of follow-up and financial constraints prevented testing different probiotic doses or prebiotic combinations, limiting insights into dose–response or synergistic effects. Differences in baseline concentrations of certain key metabolites, such as 2-(4-hydroxyphenyl)propionic acid, may confound the interpretation of intervention effects. Although statistical models accounted for time and group interactions, the presence of baseline imbalance reduces the strength of causal conclusions and should be addressed in future studies through stratified randomization or covariate adjustment. The absence of symptom improvement may be due to microbiota variability, low dosage, or short intervention. The lack of symptom improvement may have been influenced by the low baseline severity of gastrointestinal symptoms (for example, compared to marathon runners or individuals with inflammatory bowel disease) or by insufficiently accurate reporting of dietary intake and its fluctuations throughout the intervention period. Future research should explore personalized probiotic strategies and targeted metabolomics to better connect microbial changes with clinical outcomes.

## 4. Materials and Methods

### 4.1. Study Group

The study involved female dancers aged 18–36, training over 8 h weekly. The exclusion criteria included recent injuries, probiotic/prebiotic use, hospitalization, travel to tropical regions, or use of antibiotics, cannabis, or anabolic steroids within three months. Recruitment took place between October 2022 and March 2023 at an academic institution specializing in physical education, with additional efforts made at local dance schools and theaters. The study included female dance students from the Academy of Physical Education and was conducted during the busiest period of the semester with the highest academic workload. Of the 51 volunteers, 26 met inclusion criteria and were randomized (block size 4; www.randomizer.org, accessed on 22 September 2022) by the probiotic manufacturer. The double-blind design ensured that neither the participants nor the investigators knew group assignments, with allocation known only to the manufacturer. Although the participants were carefully selected based on body composition and physical activity criteria, the intended group size could not be reached. Beyond the dropout reasons outlined below, recruitment was hindered by several factors, including the very low number of male volunteers (only two expressed interest), late wake-up times caused by evening rehearsals and performances that conflicted with morning sample collection, challenges in coordinating fecal sample delivery for parallel analyses, and difficulties in ensuring dietary consistency across the intervention period. Ultimately, university participants were the only group able to meet study demands, as their schedules—often spanning 8 AM to 8 PM—included academic courses, physical training, choreography, and teaching practice. Some also performed in theaters, adding weekend workload. These conditions contributed to significant physical and psychological strain during the study period. Six participants withdrew: three without explanation and three due to missed sample or supplement collection. An additional three were excluded due to incomplete data or protocol deviations. One participant was removed as a BMI outlier. The final analysis included 16 participants: 11 in the placebo group and 5 in the probiotic group.

The probiotic group received *Lactobacillus helveticus* R0052 (CNCM I-1722) and *Bifidobacterium longum* R0175 (CNCM I-3470) (Sanprobi^®^ Stress; Sanprobi sp. z o.o. sp.k., Szczecin, Poland; lyophilisate by Lallemand Health Solutions, Mirabel, Quebec, Canada) at a daily dose of 3 × 10^9^ colony-forming units (CFU) / active fluorescent units (AFU) per capsule for 12 weeks. Placebo capsules, identical in appearance, contained maltodextrin and cornstarch. Both interventions were identically packaged and labeled only with participant numbers and instructions. To assess the participants’ adherence to the probiotic or placebo regimen, they were instructed to record their daily capsule intake and to bring back any unused capsules during the final visit. Capsule counts were used to verify compliance. All the participants completed the intervention as planned, with full dose consumption over the 12-week (84-day) period, indicating 100% adherence. All the participants gave written informed consent. Study procedures followed the Declaration of Helsinki and were approved by the Bioethics Committee (Poznan University of Medical Sciences) with approval number. 412/22. The trial was registered at ClinicalTrials.gov (NCT05567653—Effects of Probiotics on Gut Microbiota, Endocannabinoid and Immune Activation and Symptoms of Fatigue in Dancers) and reported in line with CONSORT (Consolidated Standards of Reporting Trials; Centre for Statistics in Medicine (CSM), NDORMS, University of Oxford, Oxford, UK) 2010, with the participant flow shown in Figure 4. The participants were fully informed of the study’s aims, procedures, risks, and right to withdraw. No adverse effects occurred, and no participant discontinued due to tolerability, confirming both products were safe and well-tolerated. The protocol allowed for early termination if significant dropout had occurred.

All the procedures related to sample collection and data acquisition were conducted at Poznan University of Physical Education in accordance with the established research protocols and ethics guidelines, using facilities equipped with freezers suitable for proper sample storage. One week before blood sampling, the participants underwent anthropometric measurements and provided dietary data. During an information session, they were instructed to maintain habitual dietary habits, report any changes, and notify investigators of new exclusion criteria or adverse events. They were also asked to arrive for stool collection rested, after a light evening meal and 2–3 days without physical training. Stool samples (10–30 g) were self-collected using non-sterile kits, frozen immediately at −18 °C in the participants’ home freezers, and then transferred to a −80 °C laboratory freezer for long-term storage. Samples were transported on dry ice to the analytical facility to maintain frozen conditions and ensure sample integrity.

### 4.2. Methods

To assess the probiotic’s impact on the gut metabolome, fecal samples were analyzed using ultra-high-performance liquid chromatography (UHPLC) with mass spectrometry (MS): 500 µL of a mixture of methanol, water, and acetonitrile in the proportions of 50:25:25 (*v*/*v*/*v*) with the addition of deuterated internal standards was added to 60 mg of feces. The samples were shaken at 2000 rpm at 4 °C for 30 min to dissolve the metabolites in the solution and precipitate the protein. In the next step, the samples were centrifuged for 4 min at a speed of 4000 rpm and at a temperature of 4 °C. After the samples were centrifuged, the supernatant was transferred to chromatography tubes through a 0.22 μm syringe filter. The prepared samples were analyzed on the same day using LC–MS. QC samples were prepared by mixing test samples in equal proportions and prepared in the same way as the test samples. The analysis was carried out on an ExionLC liquid chromatograph equipped with a binary pump, an autosampler, and a column thermostat, coupled with a Triple TOF 6600+ mass spectrometer (Sciex, Framingham, MA, USA). The separation was carried out on a Phenomenex Luna^®^ Omega 1.6 μm polar C18 150 × 2.1 mm (https://www.phenomenex.com/products/luna-omega-hplc-column/luna-omega-polar-c18, accessed on 9 November 2023) column for 45 min in gradient separation. The mobile phases were as follows: Phase A—water with 10 mM ammonium acetate, Phase B—acetonitrile with 0.1% formic acid. The injection volume was 2 μL and the column temperature was 20 °C. The phase flow was 0.2 mL/min. Spectral analysis was performed in the positive ion mode with a capillary voltage of 5500 V, curtain gas (CUR)—25 psi, ion source gas 1 (GS1)—45 psi, ion source gas 2 (GS2)—60 psi, and the ion source temperature of 400 °C and in the negative ion mod with a capillary voltage of 4500 V, curtain gas (CUR)—25 psi, ion source gas 1 (GS1)—45 psi, ion source gas 2 (GS2)—60 psi, and the ion source temperature of 350 °C. The spectrometer collected spectral data in the SWATH mode in the range from 50 m/z to 850 m/z. The obtained spectral spectra were analyzed using SCIEX OS software (version 3.3.1, Framingham, MA, United States) and integrated SCIEX All-In-One HR-MS/MS, NIST, and own databases. Identification was based on the analysis of precursor ions and their fragments resulting from the breakdown of the metabolite in the collision cell. The mass error was 2 ppm for precursor and fragment ions. Metabolomic preprocessing included a reliability filter (RSD > 25%) and interquartile filtering to exclude the top 5% of outliers. Untargeted metabolomics enabled broad group comparisons and detection of early metabolic shifts without prior assumptions. Due to the exploratory nature of the study, no sample size calculation was conducted, as the aim was hypothesis generation rather than confirmation. Data were not normalized but log-transformed before analysis. Metabolomic analysis was performed using MetaboAnalystR v4.0.0 (Xia Lab, McGill University, Montréal, QC, Canada). Principal component analysis (PCA) was used to explore global variation, and ANOVA–simultaneous component analysis (ASCA) assessed intervention and time effects, with significance tested via 1000 permutations. Longitudinal analysis included both intervention (PLA vs. PRO) and time (Pre vs. Post), using mixed-effects models with Kenward–Roger approximation to evaluate metabolite changes and their interaction over time. Cross-sectional analysis was performed at baseline and at the endpoint to identify metabolites with notable fold changes between the probiotic and placebo groups. *T*-tests were then applied to determine which of these differences were statistically significant, with *p*-values reported alongside false discovery rate (FDR) corrections to account for multiple comparisons.

Nutritional status was assessed using 3-day dietary records and the 14-Item Mediterranean Diet Assessment Tool (MDAT), which evaluates adherence to a diet high in fruits, vegetables, legumes, whole grains, fish, and olive oil and low in refined grains, sweetened beverages, and trans fats. The participants completed real-time dietary records independently, later reviewed with researchers, while the MDAT was conducted via interview. Gastrointestinal function was assessed using selected Rome IV criteria questions on IBS-related symptoms (abdominal pain, postprandial fullness, diarrhea, constipation). Anthropometric measurements, including body composition, were performed using bioelectrical impedance analysis (SECA GmbH & Co., Hamburg, Germany). Physical activity was reported as weekly exercise hours, and blood morphology was analyzed to support the evaluation of nutritional status. Dietary and anthropometric assessments ensured baseline group comparability. Baseline group comparisons were conducted using the *t*-test or the Mann–Whitney U test when normality (assessed via the Shapiro–Wilk test) was not met. For the Rome IV questionnaire outcomes, two-way repeated measures ANOVA evaluated main effects of time, group, and their interaction. Statistical significance was set at *p* < 0.05, with η^2^ coefficients reported for significant results to indicate effect size. Power analysis (α = 0.05, power = 0.80) indicated that 113 participants per group would be required to detect significant differences in the Rome IV variables (G*Power). Bonferroni or FDR correction would have been applied post hoc to control for multiple comparisons. Given the ordinal nature of the questionnaire data, nonparametric tests or mixed-effects models were considered for assessing within- and between-group changes. Analyses were performed using STATISTICA 13.3 (TIBCO Software Inc., Palo Alto, CA, USA).

No trial outcomes were modified after study initiation. We confirm that there were no important changes to the methods after the commencement of the trial. The study was discontinued due to difficulties in recruiting a larger number of eligible participants.

## 5. Conclusions

Our findings indicate that *Lactobacillus helveticus* R0052 and *Bifidobacterium longum* R175 may modulate the gut metabolome, though their clinical relevance remains unclear. The small sample size and focus on dancers limit broader applicability. Baseline differences in (2RS)-2-(4-hydroxyphenyl)propionic acid between the groups complicate interpretation of post-intervention effects, and the high false discovery rate suggests a possible type I error. As such, these results are exploratory and require confirmation in larger studies. Future trials should examine 2-HPP in blood and urine and assess different doses and synbiotic combinations, especially given the absence of adverse effects. According to the Strength of Recommendation Taxonomy (SORT), this study qualifies as Level B evidence with a strength of recommendation of C due to its limited sample size and absence of significant clinical effects.

## Figures and Tables

**Figure 1 ijms-26-05823-f001:**
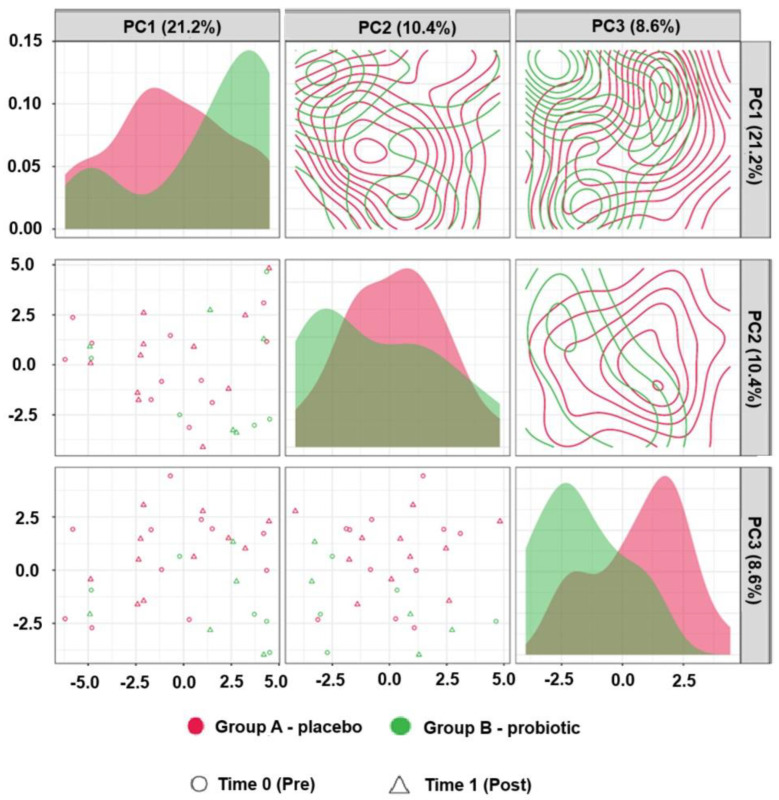
Principal component analysis (PCA) of metabolic profiles at baseline and post-intervention in the placebo and probiotic groups (A—placebo, B—probiotic). The PCA plot shows individual samples clustered by intervention group and time point.

**Figure 2 ijms-26-05823-f002:**
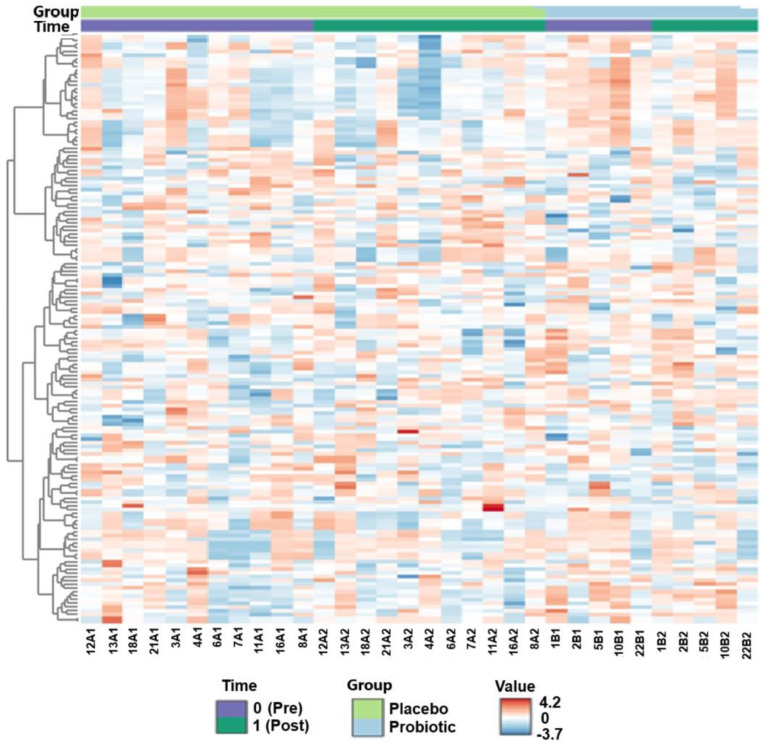
Heatmap of log-transformed metabolite values for placebo and probiotic groups (metabolite list available in Appendix A). Rows represent metabolites, columns—individual samples. Color coding shows concentration differences (red = higher, blue = lower). Clustering indicates metabolic shifts due to the intervention. The heatmap displays log-transformed (non-normalized) metabolite intensities; negative values represent low-concentration features with intensity values below 1 prior to transformation.

**Figure 3 ijms-26-05823-f003:**
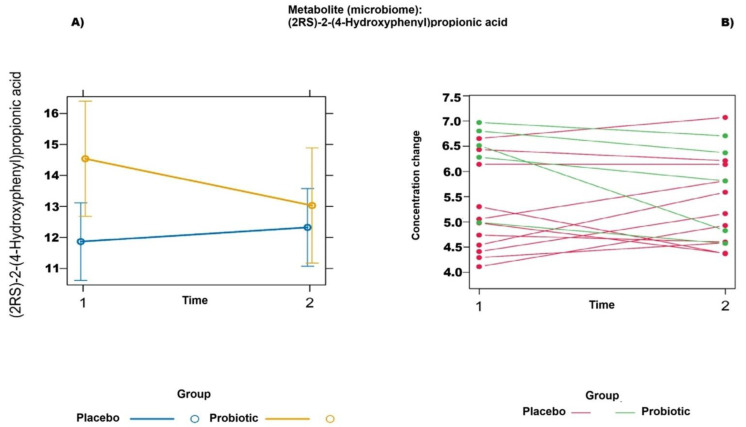
Predictor effect plot (**A**) and MEBA (**B**): intervention and time effects on (2RS)-2-(4-hydroxyphenyl)propionic acid.

**Figure 4 ijms-26-05823-f004:**
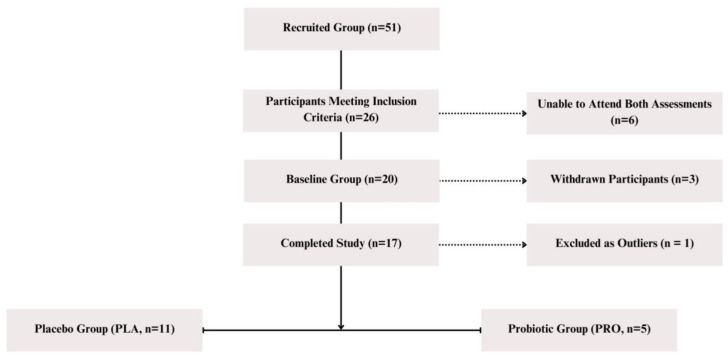
Study inclusion flow diagram. The diagram illustrates the recruitment process, inclusion and exclusion criteria, and final participant distribution into the placebo (PLA) and probiotic (PRO) groups.

**Table 1 ijms-26-05823-t001:** Baseline characteristics of the study group, including data on body mass, body composition, physical activity levels, and diet.

	Placebo (*n* = 11) Mean ± SD	Probiotic (*n* = 5) Mean ± SD	*p*-Value *
Anthropometric characteristics			
Age [years]	20.55 ± 1.04	20.00 ± 1.30	0.55 ^b^
Body mass [kg]	58.07 ± 6.95	60.10 ± 7.31	0.99 ^a^
BMI (body mass index) [kg/m^2^]	21.05 ± 2.18	20.80 ± 2.29	0.93 ^a^
Fat [% body mass]	27 ± 4	27 ± 3	0.84 ^a^
Physical activity level [hours per week]	17.11 ± 6.98	16.00 ± 9.77	0.69 ^b^
Diet composition			
Energy [kcal]	1999.23 ± 279.81	2325.54 ± 425.00	0.26 ^a^
Proteins [g]	85.29 ± 30.13	100.47 ± 21.09	0.51 ^a^
Fats [g]	74.56 ± 13.93	90.47 ± 18.72	0.41 ^a^
Total cholesterol [mg]	216.73 ± 104.36	330.21 ± 140.77	0.40 ^a^
Carbohydrates [g]	271.12 ± 53.14	298.79 ± 57.98	0.75 ^a^
Sucrose [% of carbohydrates]	10.27 ± 3.89	6.75 ± 2.81	0.09 ^a^
Fiber [g]	28.96 ± 15.13	21.36 ± 12.67	0.91 ^b^
Meditarrean diet adherence [0–14]	5.46 ± 1.86	6.20 ± 2.49	0.42 ^a^
Fruits [150–200 g portions per day]	1.64 ± 0.67	1.60 ± 0.55	1.00 ^b^
Vegetables [200 g portions per day]	2.36 ± 0.92	3.20 ± 0.84	0.10 ^b^
Legumes [150 g portions per day]	1.55 ± 1.51	2.40 ± 2.51	0.60 ^b^
Nuts [30 g portions per day]	2.46 ± 1.44	3.00 ± 1.23	0.56 ^b^
Hematologic markers			
WBC (white blood cells) [×10^9^/L]	5.54 ± 0.84	5.94 ± 1.17	0.50 ^a^
Lymphocytes [×10^9^/L]	2.34 ± 0.79	2.60 ± 0.46	0.41 ^a^

Note: * probability under the null hypothesis obtained using *t*-test or Mann–Whitney U test if normality was not assumed, ^a^ *t*-test, ^b^ Mann–Whitney U test.

**Table 2 ijms-26-05823-t002:** Cross-sectional analysis at the endpoint showing metabolites with altered concentrations in the probiotic group compared to baseline (log_2_ fold change > 2).

Metabolite	Fold Change	log_2_(FC)
Hippuric acid [M-H]^−^	8938.800000	13.1260
Curcumin (NIST EL) [M+H]^+^	354.050000	8.4678
Hyocholic acid (microbiome) [M-H]^−^	0.036575	−4.7730
Quinic acid [M+AcO-H]^−^	23.103000	4.5300
Genistein [M-H]^−^	21.530000	4.4283
Estriol [M-H]^−^	18.119000	4.1795
Naringenin (microbiome) [M-H]^−^	12.869000	3.6858
Taurine [M-H]^−^	0.119930	−3.0597
Alpha-linolenic acid [M-H]^−^	6.776500	2.7605
Alpha-linolenic acid [M+FA-H]^−^	6.574100	2.7168
3a,12a-dihydroxy-7-oxo-5b-cholan-24-oic acid (microbiome) [M-H]^−^	0.172040	−2.5392
Peimine [M+H]^+^	5.497000	2.4586
Kaurenoic acid [M-H]^−^	0.202690	−2.3027
4-Methyl-5-thiazoleethanol (microbiome) [M+H]^+^	4.839600	2.2749
Cholic acid [M-H]^−^	0.222740	−2.1666
Valproic acid [M-H]^−^	4.270200	2.0943

**Table 3 ijms-26-05823-t003:** Metabolites with significant differences at the endpoint (*t*-test results). The *t*-statistic shows the direction and magnitude of the change, with negative values indicating lower levels in the probiotic group. The *p*-value represents significance before correction, while the false discovery rate (FDR) adjusts for multiple comparisons.

Metabolite	*t*-Test	*p*-Value	−log_10_ (*p*)	FDR
Hyocholic acid (microbiome) [M-H^]−^	−2.8552	0.012719	1.8955	0.45892
L-phenylalanine [M-H]^−^	−2.7714	0.015002	1.8239	0.45892
L-tryptophan [M-H]^−^	−2.6018	0.020902	1.6798	0.45892
1-Aminocyclohexanecarboxylic acid (NIST EL) [M+H]^+^	2.5913	0.021336	1.6709	0.45892
L-phenylalanine [M+H]^+^	−2.4408	0.028545	1.5445	0.45892
3a,12a-dihydroxy-7-oxo-5b-cholan-24-oic acid (microbiome) [M-H]^−^	−2.4157	0.029951	1.5236	0.45892
Kaurenoic acid [M-H]^−^	−2.3055	0.036961	1.4323	0.45892
D-citramalic acid lithium salt (microbiome) [M-H]^−^	2.2485	0.041167	1.3855	0.45892
5-Aminopentanoic acid [M-H]^−^	−2.1452	0.049965	1.3013	0.45892

**Table 4 ijms-26-05823-t004:** Gastrointestinal complaints assessment before (Pre) and after (Post) the intervention (12 weeks between Pre and Post).

	PLA (*n* = 11) Mean ± SD		PRO (*n* = 5) Mean ± SD		2-Way ANOVA *p*-Value (η^2^)		
	Pre	Post	Pre	Post	Group	Ttime	Group × Time
Postprandial fullness [0–10]	2.91 ± 1.38	2.32 ± 1.31	3.00 ± 1.54	3.00 ± 1.83	0.508 (0.032)	0.592 (0.021)	0.592 (0.021)
Abdominal pain [0–10]	5.00 ± 1.55	4.45 ± 1.37	4.00 ± 1.58	3.80 ± 1.79	0.308 (0.074)	0.182 (0.124)	0.525 (0.294)
Menstrual pain [0–10]	7.27 ± 4.05	5.55 ± 3.21	6.80 ± 4.21	5.60 ± 4.28	0.905 (0.001)	0.224 (0.104)	0.822 (0.004)
Constipation [0–10]	4.16 ± 1.39	3.36 ± 1.67	3.92 ± 0.95	2.84 ± 1.85	0.603 (0.020)	0.030 (0.295)	0.724 (0.009)
Diarrhea [0–10]	3.64 ± 2.01	2.73 ± 1.42	2.40 ± 1.67	2.20 ± 0.84	0.262 (0.089)	0.242 (0.096)	0.448 (0.042)

## Data Availability

All data generated or analyzed during this study are included in this published article and its Appendix A.

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
