# Peer review of "A Randomized Controlled Trial Evaluating the Effects of a Probiotic Containing *Lactobacillus helveticus* R0052 and *Bifidobacterium longum* R0175 on Gastrointestinal Symptoms and Metabolomic Profiles in Female Dancers"

_ijms, 2025, doi:10.3390/ijms26125823_

Round 1

Reviewer 1 Report

Comments and Suggestions for Authors

The manuscript presents an interesting and relevant study on the effects of probiotic supplementation (Lactobacillus helveticus R0052 and Bifidobacterium longum R0175) on the fecal metabolomic profile and gastrointestinal symptoms in female dancers. The rationale is well-supported by existing literature, linking dancers' unique physiological demands with gut microbiota and metabolomic alterations. However, several areas require clarification, and some parts need further discussion.

  1. The sample size is too small (probiotic group *n*=5, placebo group *n*=11), resulting in insufficient statistical power, particularly for metabolomics analyses that require larger cohorts to control for multiple comparisons (e.g., no significant results after FDR correction).
  2. Line60-62 The choice of helveticus R0052 and B. longum R0175 is justified based on previous mental health and sleep-related studies, but their direct relevance to dancers' metabolic and gastrointestinal challenges needs stronger justification. Provide more evidence (or references) supporting these strains’ efficacy in improving gut barrier function, reducing inflammation, or enhancing recovery in physically active populations.
  3. The untargeted metabolomics approach is appropriate for exploratory analysis, but the manuscript should clarify, including which platforms/techniques were used (e.g., LC-MS, GC-MS), how data normalization and batch effects were handled and whether any pathway or network analyses were performed to interpret metabolomic shifts.
  4. Please include a brief methodological summary (or reference) of the metabolomic workflow in the Methods section.
  5. Line 188. The 2-HPP (2-(4-Hydroxyphenyl)propionic acid) is emphasized as a key metabolite, but baseline differences between groups may confound intervention effects.
  6. Extreme fold changes (e.g., hippuric acid, log₂FC > 10) raise questions about potential technical artifacts (e.g., ion suppression or contamination).
  7. The physiological implications of metabolomic changes (e.g., 2-HPP, bile acids) are not thoroughly discussed. For example, is 2-HPP linked to athletic recovery or anti-inflammatory effects?
  8. The lack of significant improvement in gastrointestinal symptoms (Rome IV criteria) weakens the practical relevance of probiotic intervention.
  9. Clarify methodological details, such as:Chromatography conditions (column type, gradient program). Mass spectrometry parameters (ionization mode, scan range). Data preprocessing workflow (e.g., MetaboAnalystR settings).
  10. No mechanistic discussion on how probiotics might modulate the gut-microbiota-metabolite axis in athletes (e.g., SCFA production, immune regulation).
  11. Line 256 The lack of GI symptom improvement (lines 250–260) contradicts some probiotic literature. The authors attribute this to microbiota variability or low dosage, but other factors (e.g., baseline symptom severity, dietary control) should be discussed.
  12. Briefly discuss whether future studies should include male dancers or stratify by sex to assess differential effects.

Author Response

To Reviewer 1:

Thank you for your constructive comments. We clarified key limitations, expanded the rationale for strain selection, and revised methodological and statistical details as suggested. Interpretations were refined to avoid overstatement, and speculative elements were addressed in the revised Discussion. We hope the manuscript now meets your expectations.

Comments & answers:

The sample size is too small (probiotic group *n*=5, placebo group *n*=11), resulting in insufficient statistical power, particularly for metabolomics analyses that require larger cohorts to control for multiple comparisons (e.g., no significant results after FDR correction).

  • We agree that the very small sample size is a limitation of our study. In the revised manuscript, this issue is explicitly acknowledged in the Discussion, where a separate paragraph has been dedicated to outlining key limitations (page 11, line 343). Additionally, relevant information regarding recruitment challenges and final sample size has been added to the “Study Group” subsection in the Methods to clarify the reasons behind participant loss and group imbalance (page 12, line 380).

Line60-62 The choice of helveticus R0052 and B. longum R0175 is justified based on previous mental health and sleep-related studies, but their direct relevance to dancers' metabolic and gastrointestinal challenges needs stronger justification. Provide more evidence (or references) supporting these strains’ efficacy in improving gut barrier function, reducing inflammation, or enhancing recovery in physically active populations.

  • We thank the reviewer for this point. We have expanded the rationale for choosing L. helveticus R0052 and B. longum R0175 by adding references and text linking these strains to gut barrier function and inflammation. However, we were unable to identify studies directly investigating these specific strains in the context of exercise-induced physiological stress or gut permeability, and we acknowledge this gap in the literature (page 2, line 81).

The untargeted metabolomics approach is appropriate for exploratory analysis, but the manuscript should clarify, including which platforms/techniques were used (e.g., LC-MS, GC-MS), how data normalization and batch effects were handled and whether any pathway or network analyses were performed to interpret metabolomic shifts.

Please include a brief methodological summary (or reference) of the metabolomic workflow in the Methods section.

  • We have added details on the analytical platform and data preprocessing in the Methods (page 13, line 434). The Methods section includes a description of the filtering steps (removal of features with relative standard deviations >25% and interquartile range filtering to eliminate 5% of variables), followed by log transformation of the data.

Line 188. The 2-HPP (2-(4-Hydroxyphenyl)propionic acid) is emphasized as a key metabolite, but baseline differences between groups may confound intervention effects.

  • We acknowledge the reviewer’s concern regarding baseline differences in 2-(4-Hydroxyphenyl)propionic acid. Information regarding this issue is provided in the Results section alongside Figure 3, where we note that the observed group × time interaction appears to be largely driven by differences already present at baseline. To improve clarity, we have added a statement in the Discussion (Limitations subsection) acknowledging that these baseline imbalances limit the strength of causal inference (page 11, line 356).

Extreme fold changes (e.g., hippuric acid, log₂FC > 10) raise questions about potential technical artifacts (e.g., ion suppression or contamination).

  • We agree that extreme fold changes raise legitimate concerns regarding possible technical artifacts such as ion suppression or contamination. However, all procedures related to sample preparation, data acquisition, and metabolomic processing were carried out following validated protocols to minimize such risks. This included the use of quality control samples, internal standards, and preprocessing steps (e.g., RSD filtering, interquartile range filtering, and log-transformation) to reduce analytical variability and exclude unreliable features.

The physiological implications of metabolomic changes (e.g., 2-HPP, bile acids) are not thoroughly discussed. For example, is 2-HPP linked to athletic recovery or anti-inflammatory effects?

  • We appreciate the reviewer’s suggestion and have expanded the Discussion to better contextualize the physiological significance of potential metabolomic changes (page 10, line 283; page 11, line 315, 336).

The lack of significant improvement in gastrointestinal symptoms (Rome IV criteria) weakens the practical relevance of probiotic intervention.

  • Thank you for this valuable comment. An appropriate statement addressing the lack of significant improvement in gastrointestinal symptoms, as assessed by the Rome IV criteria, has been added to the Limitations section (page 12, line 362).

Clarify methodological details, such as:Chromatography conditions (column type, gradient program). Mass spectrometry parameters (ionization mode, scan range). Data preprocessing workflow (e.g., MetaboAnalystR settings).

  • The Methods now include a concise outline of the untargeted metabolomics workflow (page 13, line 434).

No mechanistic discussion on how probiotics might modulate the gut-microbiota-metabolite axis in athletes (e.g., SCFA production, immune regulation).

  • We thank the reviewer for this observation. A brief mechanistic discussion on how probiotics may influence the gut–microbiota–metabolite axis in physically active individuals was included in the Introduction (page 2, line 59). In the revised version, this section has been further expanded to better reflect current knowledge on potential mechanisms such as SCFA production, modulation of intestinal barrier function, and immune regulation, with additional references provided.

Line 256 The lack of GI symptom improvement (lines 250–260) contradicts some probiotic literature. The authors attribute this to microbiota variability or low dosage, but other factors (e.g., baseline symptom severity, dietary control) should be discussed.

  • Thank you for this insightful comment. We have now included a statement explicitly addressing the absence of significant changes in gastrointestinal symptoms based on the Rome IV assessment in Discussion (page 13, line 328).

Briefly discuss whether future studies should include male dancers or stratify by sex to assess differential effects.

We have clarified this in the Limitations. While the all-female sample was a result of recruitment feasibility, we now explicitly state that the findings may not apply to males and that sex-specific responses should be explored in future studies (page 11, line 349).

Reviewer 2 Report

Comments and Suggestions for Authors

This manuscript addresses a highly relevant and timely topic—namely, the impact of probiotic supplementation on physical performance and metabolomic profiles. The study is of particular interest as it presents clinical findings reporting a lack of positive effects of the probiotic formulation on selected aspects of well-being. Beyond gastrointestinal outcomes, the analysis also includes assessments of menstrual pain perception, which is a clinically relevant and well-documented parameter. Given the strength of these findings, the article title should reflect them more directly, as the metabolomics component appears secondary in both quality and depth of interpretation.

Major Concerns:
1. Incomplete Characterization of Metabolomics Data

The manuscript contains insufficient detail and often superficial descriptions of the metabolomic results. Terminology used throughout the manuscript, such as “abundance,” is more typical of microbiome studies and is not appropriate when discussing metabolite concentrations. Descriptions are often vague or imprecise, and some interpretations appear speculative.

Lines 113–114: The claim that "higher concentrations in the probiotic group and lower in placebo suggest an intervention effect" is unsupported. Why is it assumed that probiotics can only increase metabolite concentrations?
Figure 2: It is unclear what the "Value" represented by the red-to-blue gradient actually refers to. If this denotes metabolite concentration, why are negative values reported? This requires clarification and, potentially, correction.
Figure 3: It is unclear whether the figure shows changes in metabolite concentration or detection frequency for (2RS)-2-(4-Hydroxyphenyl)propionic acid. The trend shown (a decrease in the probiotic group) appears to contradict the earlier conclusion.
This metabolite is also not listed in Table 2, despite being described as a differentiating feature.
Table 2 includes several metabolites typically associated with dietary intake (e.g., curcumin, genistein, naringenin, peimine, kaurenoic acid), suggesting that group differences in metabolomic profiles may have been influenced by diet or supplement use. This is acknowledged, albeit indirectly, in the discussion (lines 188–224).
The formation of 2-(4-Hydroxyphenyl)propionic acid (2-HPP) is dependent on the dietary intake of isoflavonoids. This points to diet as a key variable influencing both metabolomic outcomes and reported gastrointestinal effects. A re-evaluation of the dietary questionnaires is strongly advised to verify whether participants consumed significant amounts of isoflavone-containing foods (e.g., soy products), which may explain the observed differences.

2. Inadequate Dietary Assessment

Diet is a major confounding factor in both microbiota and metabolome research. While the authors state that participants completed a dietary questionnaire, the comparative analysis between placebo and probiotic groups is limited to basic nutritional parameters (e.g., caloric intake, macronutrient distribution). There is no information about specific dietary patterns/diet or the types of food consumed.

I disagree with the interpretation in Subsection 2.2, lines 105–107. The observed differences in metabolomic profiles are likely due to high inter-individual variability, largely driven by dietary differences in consumed food products.
The description for Table 2 lacks clarity regarding which group showed increased metabolite concentrations.

3. Title and Focus

The most robust and scientifically substantiated results are presented in Subsection 2.3. As such, I recommend revising the title to highlight these outcomes, while presenting the metabolomic analysis as a secondary component.

4. Probiotic Strain Citations

Lines 60–65 in the Introduction cite effects attributed to the probiotic strains used in the study. However, many of these citations are meta-analyses or review articles and do not specifically address L. helveticus R0052 or B. longum R0175. Citations should be corrected to refer to oryginal research papers directly involving these strains.

5. Methodological Clarity

Several important methodological aspects require clarification or expansion:

The study mentions blood sampling and hematological analysis, yet no such data are presented in the results.
The sample preparation process for chromatographic analysis is inadequately described. Please specify:
Sample pretreatment steps;
Chromatographic column used;
Composition of the separation phases;
Type of mass spectrometer, software version, and the version of the metabolite identification database.
It is unclear why no normalization of metabolomics data was applied. Given the nature of biological matrices such as feces and urine, where metabolite concentrations can vary drastically due to hydration status and dilution, normalization is essential.
I recommend applying:

Unitization, which ensures equal range and varied variance across variables;
Or Standardization, which equalizes the variance of variables, thus emphasizing profile differences over absolute values. These approaches should significantly enhance the reliability of the metabolomic analysis.

6. Gender Limitation

The study's findings are based exclusively on female participants. This important limitation should be explicitly acknowledged in the manuscript.

Conclusion:

This study addresses a valuable and underexplored topic. However, the current version of the manuscript suffers from major weaknesses in metabolomic data interpretation, dietary control, and methodological reporting. I recommend major revision. In particular, dietary variables must be re-evaluated, figures and tables clarified, and metabolomic analyses normalized to ensure accurate and meaningful conclusions. The title should also be revised to reflect the study’s most valid and evidence-based findings.

Author Response

To Reviewer 2:

Thank you for your valuable feedback. We revised the title for clarity, corrected terminology, clarified dietary and metabolomics data, and adjusted interpretations to reduce speculation. Methodological descriptions were also expanded. We appreciate your input and hope the current version addresses all concerns.

Comments & answers:

Beyond gastrointestinal outcomes, the analysis also includes assessments of menstrual pain perception, which is a clinically relevant and well-documented parameter. Given the strength of these findings, the article title should reflect them more directly, as the metabolomics component appears secondary in both quality and depth of interpretation.

  • We thank the reviewer for the suggestion. The title has been revised to better reflect the main findings from Subsection 2.3.

Major Concerns:

  1. Incomplete Characterization of Metabolomics Data

The manuscript contains insufficient detail and often superficial descriptions of the metabolomic results. Terminology used throughout the manuscript, such as “abundance,” is more typical of microbiome studies and is not appropriate when discussing metabolite concentrations.

  • We thank the reviewer for this observation. The terminology has been revised throughout the manuscript to more accurately reflect metabolomics conventions (e.g. Table 2, Figure 2).

Descriptions are often vague or imprecise, and some interpretations appear speculative.

  • We thank the reviewer for this important comment. In response, we have revised key sections of the Results and Discussion to improve clarity and reduce speculative language. Biological interpretations are now supported with mechanistic details and relevant references, particularly regarding metabolite pathways (e.g., aromatic amino acids). Where appropriate, we have added context about microbial activity and dietary contributions, and we have carefully reframed speculative statements to reflect the limitations of our data. Some of the more speculative interpretations have also been explicitly acknowledged in the Limitations section to ensure balanced and transparent discussion of the findings.

Lines 113–114: The claim that "higher concentrations in the probiotic group and lower in placebo suggest an intervention effect" is unsupported. Why is it assumed that probiotics can only increase metabolite concentrations?

  • Thank you for your insightful comment. We acknowledge that our statement in lines 113–114, suggesting that "higher concentrations in the probiotic group and lower in placebo suggest an intervention effect," was an overgeneralization. While probiotics can influence metabolite concentrations, their effects are not uniformly increases; they can also lead to decreases, depending on various factors such as the specific metabolite, probiotic strains used, and individual host responses (page 5, line 160).

Figure 2: It is unclear what the "Value" represented by the red-to-blue gradient actually refers to. If this denotes metabolite concentration, why are negative values reported? This requires clarification and, potentially, correction.

  • We thank the reviewer for this observation. Figure 2 presents log-transformed, non-normalized metabolite intensities. The presence of negative values reflects low-abundance features with raw intensity values below 1 prior to transformation—a common outcome in untargeted metabolomics workflows. This has now been clarified in the figure legend to avoid confusion.

Figure 3: It is unclear whether the figure shows changes in metabolite concentration or detection frequency for (2RS)-2-(4-Hydroxyphenyl)propionic acid. The trend shown (a decrease in the probiotic group) appears to contradict the earlier conclusion.

This metabolite is also not listed in Table 2, despite being described as a differentiating feature.

  • We thank the reviewer for this comment. Figure 3 represents log-transformed relative metabolite concentrations, not detection frequencies. The observed trend for 2-(4-Hydroxyphenyl)propionic acid reflects relative abundance changes across time points within each group. The metabolite is not listed in Table 2 because that table reports only statistically significant within-group changes for the probiotic arm. Since this compound did not meet the significance threshold in that specific analysis, it was not included there, although its pattern was discussed due to biological relevance. The earlier conclusion has been clarified accordingly.

Table 2 includes several metabolites typically associated with dietary intake (e.g., curcumin, genistein, naringenin, peimine, kaurenoic acid), suggesting that group differences in metabolomic profiles may have been influenced by diet or supplement use. This is acknowledged, albeit indirectly, in the discussion (lines 188–224).

  • We thank the reviewer for this observation. To address this point, we have added a clarification in the Results section noting that although several phytochemical metabolites (e.g., curcumin, naringenin) were detected, no participant reported using related supplements in the dietary logs. Additionally, dietary pattern assessment based on Mediterranean diet adherence did not indicate significant differences between groups, suggesting that these compounds may originate from unreported or habitual dietary sources rather than supplementation (page 8, line 209).

The formation of 2-(4-Hydroxyphenyl)propionic acid (2-HPP) is dependent on the dietary intake of isoflavonoids. This points to diet as a key variable influencing both metabolomic outcomes and reported gastrointestinal effects. A re-evaluation of the dietary questionnaires is strongly advised to verify whether participants consumed significant amounts of isoflavone-containing foods (e.g., soy products), which may explain the observed differences.

  • We thank the reviewer for this important comment. As noted in the manuscript, the dietary pattern assessed referred specifically to adherence to the Mediterranean diet. To address the concern, we have revised the text to clarify this and now include the average intake levels of selected food groups derived from the dietary questionnaire, providing a more detailed overview of participants’ habitual food consumption (Table 1).

  1. Inadequate Dietary Assessment

Diet is a major confounding factor in both microbiota and metabolome research. While the authors state that participants completed a dietary questionnaire, the comparative analysis between placebo and probiotic groups is limited to basic nutritional parameters (e.g., caloric intake, macronutrient distribution). There is no information about specific dietary patterns/diet or the types of food consumed.

  • We thank the reviewer for this important comment. As previously noted, dietary assessment focused on adherence to the Mediterranean diet. To address this point, the text was revised to clarify the dietary pattern evaluated, and additional data on average intake of selected food groups were included to better characterize participants’ habitual diet (Table 1).

I disagree with the interpretation in Subsection 2.2, lines 105–107. The observed differences in metabolomic profiles are likely due to high inter-individual variability, largely driven by dietary differences in consumed food products.

  • We appreciate the reviewer’s comment and recognize the relevance of inter-individual and dietary variability in metabolomic studies. In response, we have revised Subsection 2.2 to adopt more cautious language. While multiple statistical approaches (mixed-effect models, ASCA, cross-sectional analysis) indicated consistent group-level differences, and dietary records showed no major between-group variation, we acknowledge that unreported dietary factors may still have contributed to the observed patterns (page 4, line 145, line 160).

The description for Table 2 lacks clarity regarding which group showed increased metabolite concentrations.

  • We thank the reviewer for this comment. The title of Table 2 has been revised to clearly indicate that it refers specifically to within-group changes observed in the probiotic group.

  1. Title and Focus

The most robust and scientifically substantiated results are presented in Subsection 2.3. As such, I recommend revising the title to highlight these outcomes, while presenting the metabolomic analysis as a secondary component.

  • We thank the reviewer for the suggestion. The title has been revised to better reflect the main findings from Subsection 2.3.

  1. Probiotic Strain Citations

Lines 60–65 in the Introduction cite effects attributed to the probiotic strains used in the study. However, many of these citations are meta-analyses or review articles and do not specifically address L. helveticus R0052 or B. longum R0175. Citations should be corrected to refer to oryginal research papers directly involving these strains.

  • We thank the reviewer for this point. We have expanded the rationale for choosing L. helveticus R0052 and B. longum R0175 by adding references and text linking these strains to gut barrier function and inflammation. However, we were unable to identify studies directly investigating these specific strains in the context of exercise-induced physiological stress or gut permeability, and we acknowledge this gap in the literature (page 2, line 81).

  1. Methodological Clarity-Several important methodological aspects require clarification or expansion: The study mentions blood sampling and hematological analysis, yet no such data are presented in the results.
  • We thank the reviewer for this comment. Hematological indicators, specifically white blood cell count (WBC) and lymphocyte levels, have now been added to the Table 1. These parameters support the baseline comparability between groups and were included to strengthen the characterization of the study population.

The sample preparation process for chromatographic analysis is inadequately described. Please specify: Sample pretreatment steps; Chromatographic column used; Composition of the separation phases; Type of mass spectrometer, software version, and the version of the metabolite identification database. It is unclear why no normalization of metabolomics data was applied. Given the nature of biological matrices such as feces and urine, where metabolite concentrations can vary drastically due to hydration status and dilution, normalization is essential. I recommend applying: Unitization, which ensures equal range and varied variance across variables; Or Standardization, which equalizes the variance of variables, thus emphasizing profile differences over absolute values. These approaches should significantly enhance the reliability of the metabolomic analysis.

  • We have added details on the analytical platform and data preprocessing in the Methods (page 13, line 434). The Methods section includes a description of the filtering steps (removal of features with relative standard deviations >25% and interquartile range filtering to eliminate 5% of variables), followed by log transformation of the data.

  1. Gender Limitation

The study's findings are based exclusively on female participants. This important limitation should be explicitly acknowledged in the manuscript.

  • We have clarified this in the Limitations. While the all-female sample was a result of recruitment feasibility, we now explicitly state that the findings may not apply to males and that sex-specific responses should be explored in future studies (page 11, line 349).

Round 2

Reviewer 2 Report

Comments and Suggestions for Authors

The authors addressed all the comments and introduced necessary changes to the text where needed, clarified certain aspects, and provided required details in the methods. In some cases, they provided explanations of their point of view that were acceptable; therefore, no changes to the text were required.

However, in Figure 3B, the Y axis is still described as 'Abundance', consider changing to 'Relative concentration' or 'Concentration change'.

Author Response

We thank the reviewer for this thoughtful remark. In accordance with the suggestion, the Y-axis label in Figure 3B has been revised and now reads “Concentration change” to more accurately reflect the nature of the data presented.